# The Influence of Conditional Distributions on Discovered Stochastic Differential Equation Models

**Xeniya Bashkova, Alexander Hvatov**
NSS Lab
ITMO University
Saint-Peresburg, 197101, Russia
`{bashkova,alex_hvatov}@itmo.ru`

## Abstract

In today's rapidly advancing machine learning landscape, evaluating algorithm performance is paramount. As models grow more intricate, incorporating advanced mathematical frameworks like stochastic differential equations (SDEs) becomes essential for refining training processes. While SDEs inherently capture uncertainty and randomness, they also enhance predictive capabilities. This paper investigates the influence of conditional distributions within SDEs, examining how they affect the efficiency and precision of machine learning algorithms. Our analysis reveals promising avenues for developing novel strategies that could further optimize algorithm performance.

## 1 Introduction

Data-driven modeling has become inseparable from modern science and technology Solomatine et al. (2008). Compact and interpretable models such as differential equations are crucial here and allow describing various dynamic processes with high accuracy Bar-Sinai et al. (2019). However, different types of uncertainties inevitably arise when training models in the form of differential equations from observational data, where the form of the governing process equations is unknown in prior knowledge. These uncertainties related to incomplete data and poor understanding of the process can significantly affect the quality and reliability of the obtained models. Therefore, uncertainty quantification and consideration become a fundamental aspect of modeling, and its reduction becomes the most important task.

Differential equations, either ordinary differential equations (ODEs) or partial differential equations (PDEs), provide compact and information-intensive models to describe dynamic processes Rackauckas et al. (2020). Their interpretability and the possibility of modeling causal relationships make them attractive to different fields of science. In addition, it is noted that in certain cases, dynamic processes are better described by stochastic differential equations (SDEs), where stochastic effects are introduced into the model, and they model uncertainty Neklyudov et al. (2023). SDEs help capture the inherent randomness in a system and can be particularly useful in domains such as finance, biology, and climate science, where uncertainty is an intrinsic feature of the modeled process.

With the recent modification of the prominent differential equation discovery method SINDy (and PDE-FIND), E-SINDy can be interpreted as a type of SDE, allowing models to be formulated as ensembles of single (separated) ODEs or PDEs. Thereafter, they are solved to find ensemble uncertainty against the data. One step is to form a single SDE from an ensemble of ODEs/PDEs. This provides an alternative representation of dynamic systems while incorporating inherent uncertainties within the model structure. The E-SINDy approach extends traditional sparse identification of nonlinear dynamical systems by explicitly modeling the stochastic nature of the data, making it more robust to measurement noise and perturbations. This feature is particularly beneficial when working with experimental data, where noise and variability are inevitable.

Recent paper Hvatov & Titov (2023) proposes that Bayesian networks can replace marginal distributions with conditional distributions. In this case, the equation is formed as a graph with vertices

representing the equation terms conditional distribution of a given term and the edge's mutual distribution dependency. This leads to a special type of SDE, where dependencies between variables are explicitly modeled through Bayesian inference. Such an approach allows for an alternative way to incorporate uncertainty into the model while maintaining interpretability: we can sample and solve equations. Using conditional probability distributions, Bayesian networks enable a more refined representation of uncertainty, ensuring variations in one part of the system propagate appropriately throughout the model.

Comparing different types of SDE models, including those obtained through E-SINDy and Bayesian networks, presents a challenge since the models have different forms. However, the common thing that every term has something in common – the distribution of the coefficient. From this perspective, the ensemble of equations represents first-order Sobol indices, and the Bayesian network adds the second-order Sobol indices. By applying Sobol indices, we can evaluate how different modeling choices affect the stability and reliability of learned differential equations, facilitating a more informed selection of uncertainty quantification methods.

The **aim** of the work is to show how the second-order Sobol term addition affects the quality of the discovered differential equation.

The **contributions** of this work are as follows: (a) we introduce an assessment criterion using Sobol indices, which provides a systematic way to measure and compare uncertainty in differential equation discovery;
(b) we conduct a practical evaluation of the Bayesian network approach, demonstrating its effectiveness in capturing and propagating uncertainty within dynamic systems;
(c) We develop a readily available tool for uncertainty assessment in differential equation discovery, making it accessible to researchers and practitioners in various fields.

Despite our approach's advantages, certain **limitations** must be considered. These include computational complexity, as Bayesian network inference and Sobol index computations can be computationally expensive, especially for high-dimensional systems. Additionally, assumptions underlying Sobol index-based comparisons, such as independence of inputs or linearity of effects, may not always be held in complex real-world scenarios.

## 2 RELATED WORK

Several approaches can be identified in the context of discovering stochastic differential equations (SDEs) in machine learning. One common direction is using data-driven methods, as exemplified in Wang et al. (2022), where the authors employ sparse Bayesian learning to account for inherent stochasticity. However, this method is not a general one and has certain limitations. A more traditional approach is highlighted in Jacobs et al. (2023), which introduces a method based on E-SINDY Fasel et al. (2022). This framework leverages bootstrap aggregating techniques to infer governing ordinary differential equations (ODEs), which can be further extended to discovering SDEs.

The mathematical formulations of SDE solutions often reveal deep connections to partial differential equations (PDEs). These connections are rigorously examined in Karoui & Mrad (2013). Further, numerical methods often bridge the study of PDEs and SDEs, with significant contributions made in using neural networks for solving PDEs and their correlation with SDEs by Beck et al. (2019) and Beck et al. (2021). These works illustrate how deep learning methods enable efficient numerical approximations of such equations. Another notable contribution focuses on numerical solutions of PDEs and SDEs using reinforcement learning with gradient-based methods, as presented in E et al. (2017).

A separate line of study deals with neural SDEs, where novel techniques are applied to model stochastic processes using deep neural networks Kong et al. (2020) and Bayesian deep neural networks Archibald et al. (2020), Xu et al. (2022). These approaches are particularly useful for capturing complex dynamics where traditional numerical methods may fall short. Another interesting advancement in SDE solutions is SEEDS Gonzalez et al. (2024), a method praised for its ability to analytically compute the linear components of SDE solutions, including the novel use of the exponential time difference technique, which enhances efficiency and accuracy.

In addition to solving SDEs, their formulation often depends heavily on variance, highlighting the significance of variance-based methods in estimating SDE solutions. For instance, in Rhee & Glynn (2015), the authors propose a method for unbiased estimators with finite variance, achieving a square-root convergence rate. Similarly, Kim & Reich (2023) extends the concept of variance-based estimators by leveraging minimum variance techniques and their connection to deterministic control laws, thereby offering a robust framework for variance-dependent estimation problems, which is of interest.

## 3 METHODOLOGY

### 3.1 DIFFERENTIAL EQUATION DISCOVERY AT A GLANCE

The goal of the classical differential equation discovery is to get the equation in the form:

$$L = \sum_{i=1}^{i=N_{terms}} a_i t_i \tag{1}$$

, with $t_i$, we denote differential terms, i.e., the ones that contain only function and its derivatives, and with $a_i$ - algebraic terms or coefficients. Usually, it is assumed that coefficients contain functions of independent variables. For simplicity throughout the paper, we will consider only constant coefficients. The equations may be obtained by various means. We mention a few properties:

- Differential equation is an implicit model, so it cannot be compared with data directly
- The data for the equation should be additionally pre-processed to get derivatives. It also means that noise in data introduces noise in the resulting equation; for simplicity, it is taken out of consideration

The complex is an automated approach based primarily on the EPDE framework , which allows obtaining machine learning models as equations. The initial stage of data pre-processing normalizes the data to bring it to a standard format, facilitating subsequent calculations and minimizing the impact of the scale and distribution of the data on the results. The extent to which the data is of good quality and corresponds to the model's expectations is determined in the process. The algorithm parameters are set, which determine the possible set of equations. These parameters include:

1. Maximum equation order: determines what order of descriptive derivatives can be included in the equation.
2. Maximum number of terms: the maximum number of different terms that can be included in the final equation.
3. Maximum nonlinearity order: specifies which nonlinear terms are acceptable, for example, the presence of second-order terms such as or more complex ones.
4. Maximum degree of each element in the term: allows one to set limits on the degree to which elements (variables, derivatives) can be raised.
5. Basic functions: such functions as sin, cos, exp, and others are defined as additional functional expressions, which can be useful in constructing complex dependencies.

After setting up all the input parameters of the process, all subsequent steps are performed fully automatically. The algorithm uses machine learning methods to determine the parameters of the equation based on the initial data. Searching for ordinary differential equations (ODE) using the EPDE framework, a key procedure for selecting structural components of the equation, is used through regression using the LASSO (Least Absolute Shrinkage and Selection Operator) method. This method allows one to identify which terms of the equation should have non-zero coefficients. At the initial stage, the algorithm determines which terms of the equation can be significant based on normalized data. This is achieved using LASSO regression, which uses L1 regularization. Unlike conventional regression, which minimizes the sum of squared deviations, LASSO adds a penalty proportional to the modulus of the coefficients. This regularization helps reduce the impact of overfitting and

zero out some coefficients, thereby removing unnecessary terms. Each equation undergoes multi-criteria optimization, in which each equation is assigned a specific regularization coefficient, which determines the penalty for adding unnecessary terms. Optimization is carried out through LASSO regression, which, by varying the intensity of regularization, allows finding a balance between the complexity of the model and its compliance with the training data.

## 3.2 ENSEMBLE-BASED ALGORITHM

Ensembles of equations may be obtained using different approaches—either by data and pre-defined library bootstrapping, like it is done in E-SINDy, or with stochastic by nature algorithms, such as evolutionary optimization like EPDE DLGA/SGA PDE or RL algorithms like DISCOVER.

The ensemble may be treated as a "realization" or certain SDE model, so it could be approximately reversed by gathering term coefficients statistics. Thereafter, the statistics are used to determine the marginal distributions $p_i$, so the equation has the following form:

$$L = \sum_{i=1}^{i=N_{terms}} p_i t_i \tag{2}$$

We cannot check directly how marginal distributions $p_i = p(a_i|t_i)$ affect the resulting solution since the dependency is formed explicitly. We again sample equations using 2 using the Monte Carlo and solve them to show uncertainty in the data space and assess the distribution parameters of the solution.

An overview of the described approach is presented below.

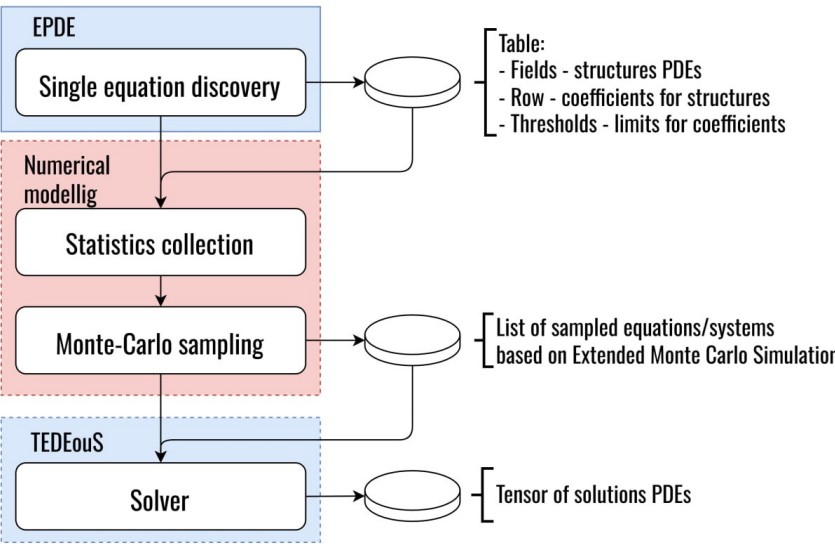

Figure 1: Description of the proposed approach.

## 3.3 EXPLORING THE BAYESIAN APPROACH

Statistics may be used in another way - to form a Bayesian network. In a Bayesian network, conditional probabilities are computed for each node, given the values of its parent nodes. Eventually, after training, the Bayesian network serves as a probabilistic model representing the dependencies between the terms of the equations, from which equations corresponding to the identified patterns can be sampled. In this case, differential equations can be considered graph models with additional strong interconnection (mutual correlation) between nodes, and the tabular structure presented is the basis for constructing a joint distribution reflecting the frequency of occurrence of the terms in the obtained equations.

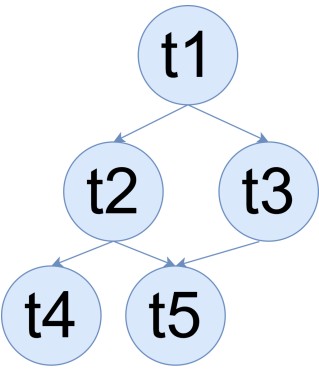

Figure 2: Caption

The illustration of the BN model is shown in Fig. 2.

The resulting equation will have the form

$$p_1 t_1 + p_2 t_2 + p_3 t_3 + p_4 t_4 + p_5 t_5 = 0 \tag{3}$$

Inherently (it is an assumption made in Bayesian network construction), we will have a single term, such as rooting, which is the initial term. In our example, the term $t_1$ has only a marginal distribution $p_1 = p(a_1|t_1)$ and is used to pass the probabilities to other terms. In particular, $p_2 = p(a_2|t_2, a_1)$, $p_3 = p(a_3|t_3, a_1)$, $p_4 = p(a_4|t_4, a_2, a_3)$ and $p_5 = p(a_5|t_5, a_2)$ (Bayesian networks usually use somewhat inverse dependency notation shown with arrows in Fig. 2). We note that some coefficients could be zero; it is not necessary that every term present in the sampled equation. Also, there may be a situation where we could form several separate components, Bayesian networks; in this case, equations are samples from all components simultaneously.

Like in the previous case, we could not directly assess the influence of every distribution on the resulting solution. Therefore, we sample and solve equations to collect statistics in the data space. The full pipeline is shown in Fig. 3.

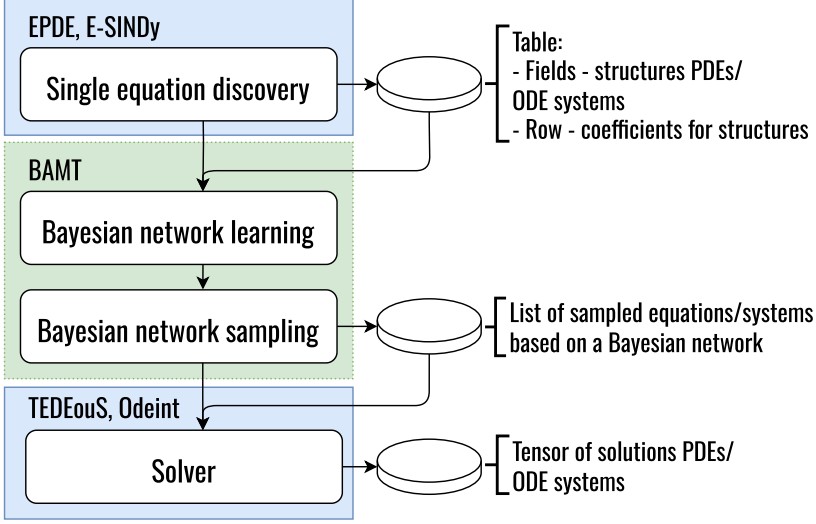

Figure 3: Universal approach for differential equation discovery.

### 3.4 SOBOL ANALYSIS

The main purpose of the research is to compare two approaches for robust training of models in differential equation form based on stochastic differential equations (SDE) and Bayesian networks (BN) and to quantify/estimate the contribution of each parameter and their combinations/interactions using the Sobol method.

The Sobol method relies on decomposing the variance of the function $f(\mathbf{x})$ into components of different orders, as shown in Eq. equation 4:

$$f(\mathbf{x}) = f_0 + \sum_{i=1}^{n} f_i(x_i) + \sum_{1 \leq i < j \leq n} f_{ij}(x_i, x_j) + \cdots$$
$$+ f_{1,2,\ldots,n}(x_1, x_2, \ldots, x_n). \quad (4)$$

Every equation obtained using either ensembles or Bayesian networks could also be decomposed using Sobol decomposition into

$$Lu \sim \underbrace{\sum_{i=1}^{i=N} c_i t_i}_{\text{"mean" equation}} + \underbrace{\sum_{i=1}^{i=N} p(a_i, t_i) t_i}_{\text{classical SDE}} +$$
$$+ \underbrace{\sum_{i=1}^{i=N} \sum_{j \neq i} p(a_i, t_i, a_j) t_i}_{\text{conditional distribution}} \quad (5)$$

In the context of our research, we focus on estimating the contribution of individual parameters and second-order interactions, which corresponds to the SDE and BN approaches in solution (data domain), respectively. We relate the components of the variance decomposition to two similar approaches of training models in the form of differential equations that differ in the degree of uncertainty treatment: SDE and BN. Eq. equation 6 shows this relationship.

$$u(\mathbf{x}, \mathbf{y}, \mathbf{t}) \sim \underbrace{f_0}_{M[u(\mathbf{x},\mathbf{y},\mathbf{t})]} + \underbrace{\sum_{i=1}^{n} f_i(x_i)}_{[SDE] \text{ and } [BN]} +$$
$$+ \underbrace{\sum_{1 \leq i < j \leq n} f_{ij}(x_i, x_j)}_{[BN]} \quad (6)$$

As a result, two methods are compared as shown in Fig. 4.

$$u[SDE] = M[SDE] + D_1[SDE]$$
$$u[BN] = M[BN] + D_1[BN] + D_2[BN]$$

Approx. equal    Approx. equal

Figure 4: Scheme of dispersion distribution of both approaches, with $D_1[\cdot]$ first-order Sobol indices are denoted and with $D_2[BN]$ - second-order Sobol index is denoted

We must assess how the second-order Sobol indices $D_2[BN]$ are compared with the first-order $D_1[SDE]$. We assume that the mean values $M[SDE]$ and $M[BN]$ and the first-order indices $D_1[SDE]$ and $D_1[BN]$ are approximately the same. Therefore, we assess the input of second-order indices (i.e. conditional distribution part influence) as

$$R2 - 1 = \frac{D[(u[SDE] - M[SDE])]}{D[(u[BN] - M[SDE])]} \tag{7}$$

Equation Eq. 7 shows the inverse $R2$ value that shows how much dispersion is described with the second-order indices $D_2[BN]$.

## 4 Experiments

Selected examples of key data: 1. Wave equation with one spatial variable 2. Burgers equation 3. Lotka-Volterra equations

### 4.1 Experimental setup

#### 4.1.1 Wave equation with one spatial variable

We use the open-source wave process data as input data for the EPDE discovery process.

For example, a comparison of the decision fields with the initial data for our models is shown. The general solution of each model is presented as a tensor of dimensions $t, 70, 70$, where $t$ is the size of the selected interval. The resulting models can be compared in terms of quality, and it can be observed that within the Sobol analysis, all solutions are in the solution field.

$$\frac{\partial^2 u}{\partial t^2} - \frac{1}{25}\frac{\partial^2 u}{\partial x^2} = 0 \tag{8}$$

The initial heat map 5 illustrates that the solution undergoes continuous alterations as the value of the function escalates, transitioning from dark to bright/yellow regions. The next heat map demonstrates a more streamlined system behavior, resembling the initial graph. This observation suggests that the mean value closely approximates the system's initial random solution. A discernible and uniform increase in the value is evident, devoid of pronounced fluctuations. The magnitude of the difference is minimal in the third heat map, as the values fluctuate around 0, with a maximum of ±1. This finding suggests a high degree of agreement between the two approaches.

The calculated metric Eq. 7 tends to 0 and has the value shown in Tab 1.

Table 1: Statistics comparison for wave equation

| Parameter | Value |
|-----------|-------|
| $R^2$ | 0.9456 |

Additional 3D visualizations of the results Fig. 9 and Fig. 10 are provided in Appendix A for clarity.

#### 4.1.2 Burgers equation

Burgers equation form stated as shown below 9.

$$\frac{\partial u}{\partial t} + u\frac{\partial u}{\partial x} = 0, \tag{9}$$

In 6 for most values of $x$, the BS and SDE methods produce similar results ($M_{BS} \approx M_{SDE}$). This is evident from the predominant purple tones observed in the graphs.

Up to $x \approx 10$, the difference between the two methods is almost negligible.

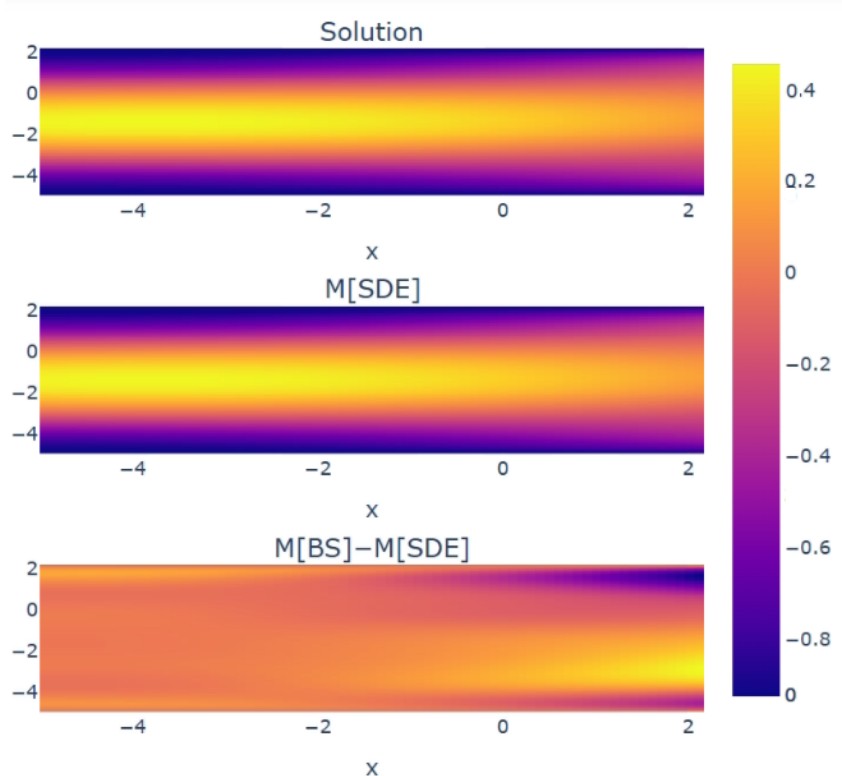

Figure 5: Basic statistical results for Wave equation.

The calculated metric 7 tends to 1 and shown in Tab. 2.

Table 2: Statistics comparison for Burgers equation

| Parameter | Value |
| --- | --- |
| $R^2$-1 | 0.96429 |

### 4.1.3 LOTKA-VOLTERRA EQUATIONS

For the EPDE discovery process, we employ a data set that illustrates the changes in the populations of lynx predators and hare prey, focusing specifically on the dynamics described by the Lotka-Volterra equations.

$$\begin{cases} \frac{du}{dt} = \alpha u - \beta uv, \\ \frac{dv}{dt} = \gamma uv - \delta v, \end{cases} \tag{10}$$

where $u$ represents the prey population and $v$ represents the predator population. The coefficients in the equations have the following interpretations:

- $\alpha$ — the coefficient of prey population growth, reflecting the natural increase of prey in the absence of predators.
- $\beta$ — the coefficient describing the decline in the prey population due to predators.
- $\gamma$ — the coefficient characterizing the predator population's growth due to prey availability.
- $\delta$ — the coefficient describing the natural decline of the predator population in the absence of prey.

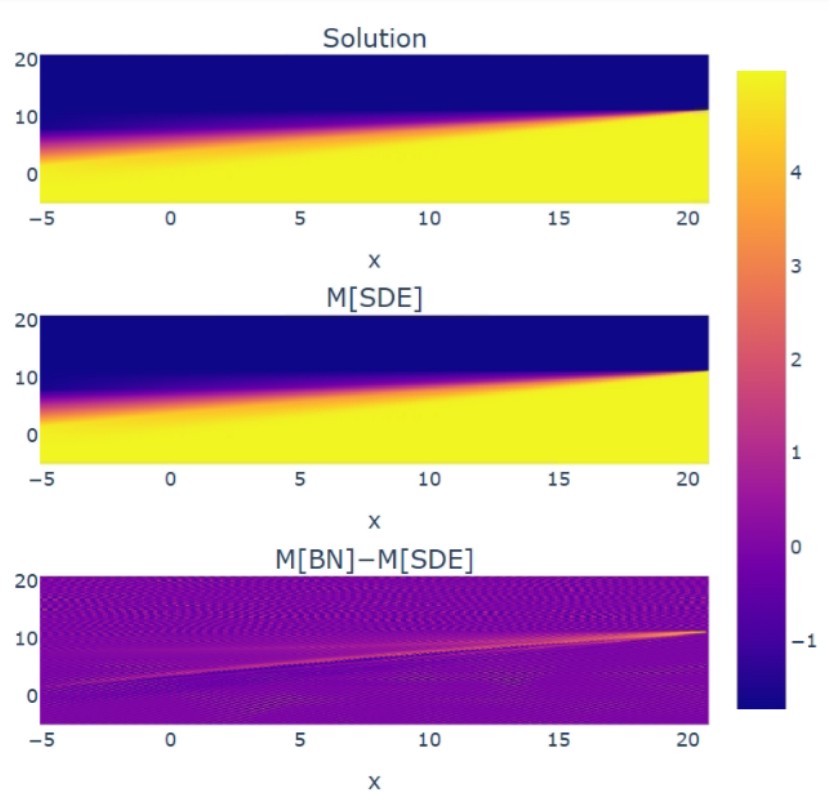

Figure 6: Provided comparison for Burgers equation.

The comparative solution is presented in two figures for preys 7 and predators 8.

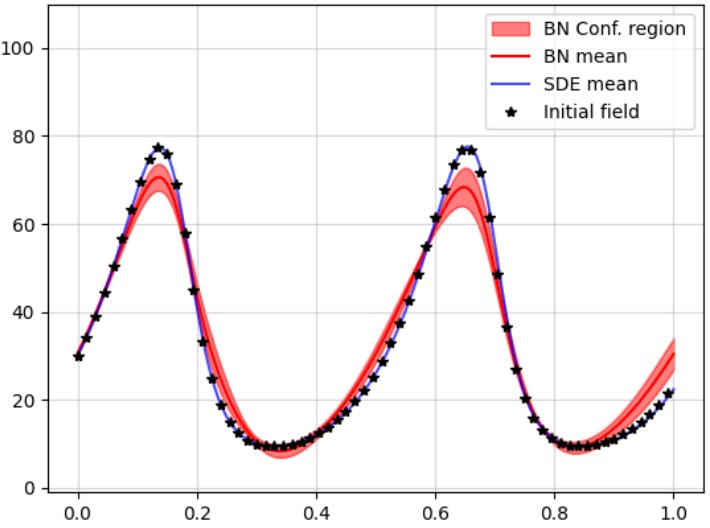

Figure 7: Numerical comparison results for the Lotka-Volterra equation for prey-hares.

The calculated metric Eq. 7 is shown below in Tab. 3.

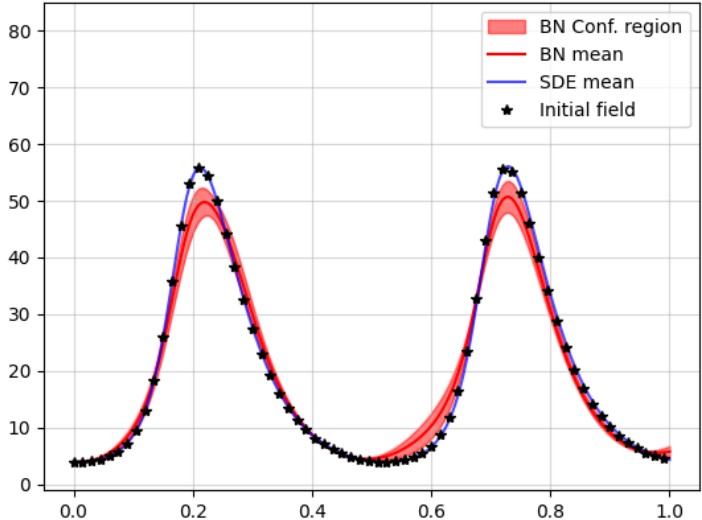

Figure 8: Numerical comparison results for the Lotka-Volterra equation, namely for predators-lynx.

Table 3: Statistics comparison for Lotka-Volterra

| Parameter | Value |
|---|---|
| $R^2 - 1(\text{hares})$ | 0.7325 |
| $R^2 - 1(\text{lynx})$ | 0.8216 |

## 5 DISCUSSION

For simple applications, the impact of second-order Sobol indices is at the low threshold (around 5 %). However, the impact increases for the real data, as shown in the Lotka-Volterra system. Therefore, for simple applications, ensemble models are good approximations of dispersion, whereas for the more complex, second-order Sobol indices are more significant.

## 6 CONCLUSION

The paper shows how equation discovery may assess uncertainty based on ensembling. The main conclusions are:

- We may assess the dispersion automatically, using gathered equation statistics either simply by ensembling the equation or by building the Bayesian networks
- Second-order Sobol indices may impact the dispersion at a relatively high level (around 20 %)
- Simple ensembling, however, is a good approximation and still preserves 80 % of the total dispersion

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

## A APPENDIX

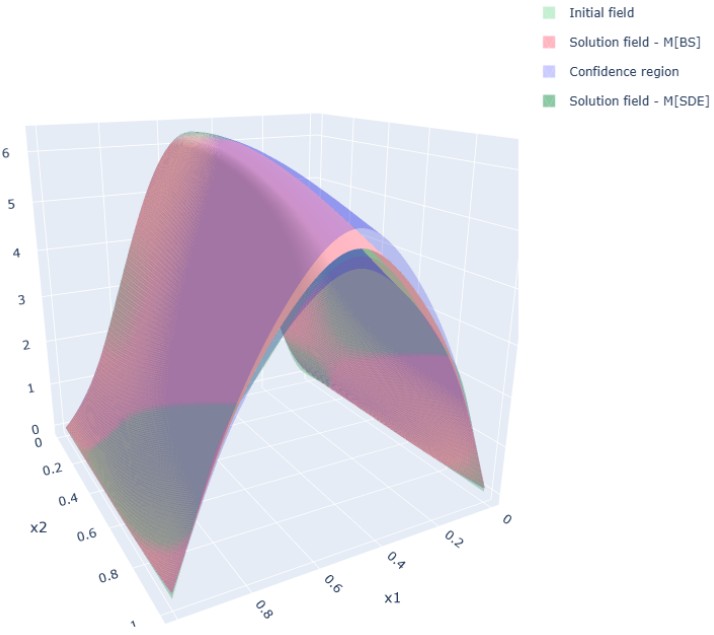

Figure 9: Visualization of the wave equation. The use of volumetric visualization allowed us to compare the two approaches, thereby showing that all solutions are in the solution domain.

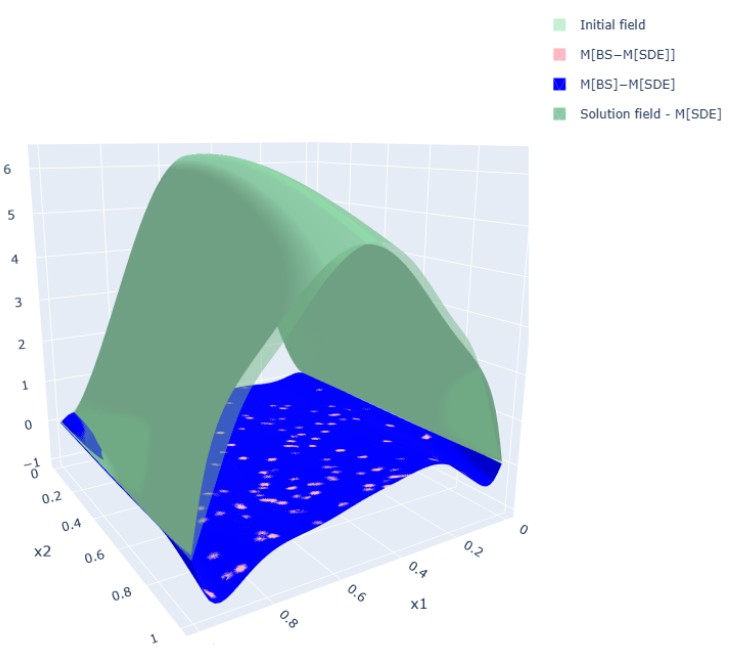

Figure 10: Visualization of the wave equation. A notable observation is the presence of a stationary process as a designation of the difference $M[BN] - M[SDE]$.

