# OpenReview forum: "The Influence of Conditional Distributions on Discovered Stochastic Differential Equation Models"
_mathai.club/MathAI/2025/Conference — MathAI 2025 Oral_

### Official Review · Reviewer_fooh · 2025-02-25
**The review highlights the innovative methodology and practical utility of the paper in quantifying uncertainty in SDE models, while noting validation gaps and technical assumptions that need addressing for broader impact.**

**Rating:** 7
**Confidence:** 4

**Review:**

The paper examines how conditional distributions affect machine learning models based on stochastic differential equations (SDEs). The authors propose a method using Sobol indices and Bayesian networks to quantify uncertainty in SDE models, improving upon traditional approaches. The study emphasizes the role of second-order interactions in uncertainty analysis, particularly in systems like the Lotka-Volterra model, where these interactions account for ~20% of variability.

The work introduces a systematic framework for uncertainty quantification, including a new metric for evaluating second-order Sobol indices. However, methodological details such as training dataset sizes, hyperparameter choices, and computational costs remain unclear. While synthetic benchmarks demonstrate potential, the lack of validation on real-world data limits insights into the method's robustness. The paper's technical assumptions and jargon-heavy explanations may hinder accessibility for interdisciplinary audiences. Addressing these issues would strengthen the paper's impact and reproducibility.

Overall, the paper offers a new and useful framework for measuring uncertainty in SDE models. While it has some limitations, particularly in validation and technical assumptions, it provides valuable insights and tools for researchers.

---

### Official Review · Reviewer_84D8 · 2025-02-25
**Review of 'The Influence of Conditional Distributions on Discovered Stochastic Differential Equation Models'**

**Rating:** 7
**Confidence:** 3

**Review:**

__Summary__:
The paper investigates the comparison between stochastic differential equations (SDEs) and Bayesian networks (BNs) constructed using ensemble-based algorithms for differential equation discovery. The focus is on assessing how these models capture and quantify uncertainty through the computation of second-order Sobol indices.

__Evaluation__:
While the paper presents an interesting approach, several key aspects need further clarification. The methodology behind training the Bayesian network used in comparisons is not fully detailed, e.g. size of the data used for ensembles, other hyperparameters. The computational cost of Bayesian network inference and Sobol index computation is mentioned in the introduction but not adequately quantified or addressed in the experiments.

__Clarity__:
The paper has a few formatting inconsistencies, such as the unexplained "BS" in the figure, which I assume refers to Bayesian Networks or the inverse of R² and some other. While these are minor, providing definitions for such terms and addressing small formatting issues could help improve clarity.

---

### Official Review · Reviewer_uCqz · 2025-02-27
**The review of The Influence of Conditional Distributions on Discovered Stochastic Differential Equation Models**

**Rating:** 7
**Confidence:** 3

**Review:**

The work considers the practical point of view at comparison of SDE and Bayesian networks based models by estimation of Sobol indices. The main result is alternative way of estimation of second-order Sobol indices for the SDE and Bayesian networks pair of models.

The basis of the estimation is given in the Figure 4, that states approximate equality of several values, what is very plausible and is supported with practical experiments, however it is not supported with any proof or reference.

The work has couple of minor clarity drawbacks in Figures 5 and 6, the axes are labeled as 'x' only, but the solution is dependent on 'x' and 't'; Figure 5 has label 'M[BS]', however 'BS' was not mentioned, and the difference in 'M[BN]-M[SDE]' figures could be more informative, if some numerical norms of this figures were presented.
The obtained numerical results show fine approximation of solution, however the authors do not present the discovered equations, thus it is better to clarify whether the aim is to assess the uncertainties in solution approximation or in equation discovery.

Overall, the paper gives an introduction to noval approach in estimation of second order Sobol indices, but it lacks strict derivation and limitations.

---

### Decision · Program_Chairs · 2025-03-08

**Decision:**

Accept (Oral)

**Comment:**

Your article has been accepted and you can give a talk on the article. All articles will be sorted by rating and within the available conference places one author from each article will be invited. If there are not enough places, then you will either have the opportunity to speak remotely or come at your own expense!